# Electrocatalytic Performance of Ethanol Oxidation on Ni and Ni/Pd Surface-Decorated Porous Structures Obtained by Molten Salts Deposition/Dissolution of Al-Ni Alloys

**DOI:** 10.3390/ijms24043836

**Published:** 2023-02-14

**Authors:** Dawid Kutyła, Kano Nakajima, Michihisa Fukumoto, Marek Wojnicki, Karolina Kołczyk-Siedlecka

**Affiliations:** 1Faculty of Non-Ferrous Metals, AGH University of Science and Technology in Kraków, Mickiewicza 30 Ave., 30-059 Krakow, Poland; 2Department of Materials Science, Graduate School of Engineering Science, Akita University, Akita 010-8502, Japan

**Keywords:** porous Ni, molten salts electrodeposition, galvanic displacement reaction, ethanol oxidation

## Abstract

Ni coatings with high catalytic efficiency were synthesised in this work, obtained by increasing the active surface and modifying Pd as a noble metal. Porous Ni foam electrodes were obtained by electrodeposition of Al on a nickel substrate. Deposition of Al was carried out with potential −1.9 V for a time of 60 min in NaCl–KCl-3.5 mol%AlF_3_ molten salt mixture at 900 °C, which is connected with the formation of the Al-Ni phase in the solid state. Dissolution of Al and Al-Ni phases was performed by application of the potential −0.5 V, which provided the porous layer formation. The obtained porous material was compared to flat Ni plates in terms of electrocatalytic properties for ethanol oxidation in alkaline solutions. Cyclic voltammetry measurements in the non-Faradaic region revealed the improvement in morphology development for Ni foams, with an active surface area 5.5-times more developed than flat Ni electrodes. The catalytic activity was improved by the galvanic displacement process of Pd(II) ions from dilute chloride solutions (1 mM) at different times. In cyclic voltammetry scans, the highest catalytic activity was registered for porous Ni/Pd decorated at 60 min, where the maximum oxidation peak for 1 M ethanol achieved +393 mA cm^−2^ compared to the porous unmodified Ni electrode at +152 mA cm^−2^ and flat Ni at +55 mA cm^−2^. Chronoamperometric measurements in ethanol oxidation showed that porous electrodes were characterised by higher catalytic activity than flat electrodes. In addition, applying a thin layer of precious metal on the surface of nickel increased the recorded anode current density associated with the electrochemical oxidation process. The highest activity was recorded for porous coatings after modification in a solution containing palladium ions, obtaining a current density value of about 55 mA cm^−2^, and for a flat unmodified electrode, only 5 mA cm^−2^ after 1800 s.

## 1. Introduction

Porous Raney nickel is one of the industrial practices’ most popular used materials. In its almost 100-year history, it has been used as a catalyst in a considerable number of inorganic reactions as well as complex organic compounds [1,2,3]. Its wide application is related to the very high versatility of this material in the chemical industry and its relatively low price and ease of preparation [4]. The Raney nickel synthesis process is based on the metallurgical synthesis of Ni-based binary alloys [5,6]. After the casting process, it is subjected to the chemical rinsing of zinc, aluminium or silicon. Their susceptibility dictates the choice of these elements to concentrated alkali, which selectively removes them from the alloys, leaving a porous Ni skeleton with a large specific surface area [7]. There is significant research on the influence of the Raney nickel synthesis conditions on the degree of surface development and its subsequent use in catalytic processes [8,9,10,11,12,13]. These works mainly concern optimising the metallurgical process of nickel-based alloy preparation and its subsequent leaching in various alkali-based solutions. Another way to synthesise this material is to use electrochemical processes in molten salt solutions based on chlorides and fluorides. This method, described in the works published by Fukumoto, is based on the electrochemical deposition of metallic Al from high-temperature molten salts, where the nickel sheet plays the role of the cathode in the system [14,15]. During the growth of the aluminium layer, the deposited layer penetrates the nickel sheet and the Al-Ni phases form, which is a spontaneous process. In the last stage, the obtained material is electrochemically dissolved by the positive polarity of the working electrode. Researchers have observed that the morphology of the obtained porous structures can be controlled by the synthesis temperature, the duration of the synthesis and the dissolution potential of aluminium and the intermetallic phase [14]. This process can synthesise single-metal sponges based on nickel and its alloys with cobalt [16]. The materials prepared in this way can serve as stable and highly active electrodes in water decomposition in an alkaline environment. In recent years, an increase in work on the electrolytic oxidation of ethanol has been observed, which is associated with a significant increase in the efficiency of fuel cells. One of the most studied elements dedicated to this type of solution is nickel. This is because during the oxidation process, nickel creates an electrochemically active redox pair Ni(OH)_2_/NiOOH, for which the oxidation reaction has a reduced overpotential [17]. Currently, the main trends in this field are the synthesis of nickel-based alloy materials, such as NiOOH-CuO oxide nanostructures [18] NiFeOOH [19], metal-modified Ni nanowires precious metals such as Rh [20] and porous nickel electrodes [21,22,23,24]. One of the most commonly used surface modifiers are palladium-based nanoparticles due to their highest activity in the oxidation of ethanol to acetate [25,26,27].

The novelty of this study is connected with the spontaneous process of surface modification of a nickel sponge by an exchange reaction. A thin layer of metal palladium was applied to the nickel surface by dipping the nickel sponge into an acidic solution containing low-concentration Pd^2+^ ions. The nickel substrate used in this work was synthesised in molten salts electrolysis/dissolution, which created the porous electrode. The proposed method of obtaining porous materials by selective dissolution in molten salts has yet to be extensively described in the scientific literature. However, this process, due to many variable parameters, such as the molten salt composition, the substrate material and the deposited metal selection, or the parameters of electrochemical deposition and dissolution of the intermetallic phase, allows for precise control of the obtained porous metallic structure. According to our knowledge, this is the first work based on molten salt electrolysis/dissolution electrode preparation dedicated to ethanol oxidation reaction. The influence of the surface development of nickel electrodes on the electrochemical oxidation of ethanol based on a flat nickel electrode and a porous sponge was investigated. Similar tests were carried out for nickel materials modified with palladium ions at different time. 

## 2. Results and Discussion

### 2.1. Electrodeposition and Structural Characterisation of the Porous Ni Electrodes

The schematic model of porous structure formation in the process of the molten salt is presented in Figure 1.

Based on Fukumoto and colleagues’ other works, the surface development level is strongly connected with selected deposition/dissolution potentials and the temperature of the molten salt electrolyte during the formation of alloys [14,15,16]. Deposition of Al on Ni substrates in molten salts can be a promising method for multiplying the electrochemically active surface area. The mechanism of the porous structure is related to the reaction between electrodeposited aluminium and nickel plate. Plated Al under the applied temperature is in the liquid state, which increased the reaction rate between these two elements. Porous structure formation was a consequence of the potential change toward more positive values, which allowed for the dissolution of the deposited Al-Ni phase. The electrochemical measurements during the deposition/dissolution process are presented in Figure 1.

The registered current density during the potentiostatic porous layer formation is presented in Figure 2. The left part of the curve was connected with Al deposition, which was performed by the application of −1.9 V, and corresponded to observing the cathodic current density for 3600 s. After this time, the potential value was modified to −0.5 V, which changed the registered current density to more positive values. Different stages can be described during the dissolution due to significant differences in the registered current density. Immediately with the application of a more positive value, the freshly deposited Al was dissolved (range 3600–4000 s). A visible drop of the anodic current density was registered due to the beginning of the Al-Ni phase dissolution. According to the thermodynamical data and binary diagram of the Ni-Al system, the solid-state reaction between nickel and aluminium can form intermetallic compounds with different Al-Ni ratios [28]. In time frameworks between 4500 and 6000 s, the linear diminishing of the registered current density has been observed, which is connected to the diffusional character of Al-Ni phase dissolution. After this time, the current density value drop is observed, and the anodic current diminishes to zero, which indicates the end of the Al-Ni phase dissolution and the reaching of the Ni substrate. The scanning electron microscope images for flat Ni plate and porous Ni electrode after molten salt treatment are presented in Figure 3. 

In the case of the flat electrode, it was subjected to mechanical polishing, and there was no inhomogeneity on its surface. On the other hand, a nickel sponge synthesised by electrochemical deposition and digestion had visible pores, even with slight magnification. The morphology of the sponges consisted of large heterogeneous networks of rounded connections, which were intersected by smaller pores of micrometric dimensions, followed by the Al-Ni leaching process and the intermetallic phase during the anodic dissolution of the alloy. The structure’s branching was visible both on the surface and in the depths of the observed holes, which proves a high degree of porosity. The crystalline structure of flat and porous nickel electrodes is presented in Figure 4.

The diffractograms are characterised by the presence of three intense peaks, originating from different crystallographic orientations of nickel: Ni (111), Ni (200) and Ni (220), respectively, at the angle values: 44.51, 51.85 and 76.37. Reflections were identified according to the JCPDS card: Ni—00-003-0850. Based on the Sherrer equation, the size of the crystallites for each reflection was determined. The porous Ni sample was characterised by clearly larger grain sizes, respectively, of 42.79 nm, 33.05 nm and 29.47 nm for reflections (111), (200) and (220), respectively. For the flat nickel sample, these values were 24.29 nm (111), 25.01 nm (200) and 23.06 nm (220). In the case of the porous sample, no peaks from the Al-Ni phases were observed, which proves the complete dissolution of the formed intermetallic phases in the electrolysis process in molten salts.

### 2.2. Electroless Modification of Flat and Porous Ni Electrodes by Pd^2+^ Ions

Galvanic displacement occurs spontaneously when a substrate metal comes into contact with a more noble metal cation. In this case, it is thermodynamically favourable for the more noble metal cation to exchange the less noble metal according to the electrochemical series [29]. The galvanic displacement process is one example of corrosion of the template. This technique allows the surface’s decoration without destroying the substrate’s structure. The galvanic displacement process also forms a continuous layer, which is beneficial for durability in hydrogen/oxygen evolution reactions [30,31,32,33]. In this particular issue, the open-circuit potential is measured for nickel flat and porous samples, and galvanic displacement can be used for the coverage measurement of the Pd on the nickel surface. Registered potential values during the immersion in Pd ions-consisting electrolytes are presented in Figure 5.

Three distinct stages characterise the course of the potential change. The first one, related to a very sharp decrease in the potential value, is related to achieving the value of the corrosion potential in such an acidic (pH = 1) solution. Such a significant change in potential is connected with the appearance of metallic palladium on the electrode surface. The second stage is the curve flattening, gradually reaching the minimum as a function of time. It is connected with the diffusion of Pd^2+^ ions and the transport of Ni^2+^ ions from the electrode surface. For a flat electrode, this step is short and takes about 400 s due to the low surface roughness and simple shape. The decrease in the potential value is related to the decreasing amount of unreacted nickel surface. A straight line, recorded from 1200 s to the end of the measurement, indicates no change in the potential value related to the total coverage of the nickel electrode by palladium. In the case of a porous nickel electrode, stage 2 is much longer. It results from the fact that the degree of surface development controls the diffusion of palladium and nickel ions in porous materials. It was observed that after 3000 s, a potential value analogous to that of the flat electrode was reached, which may mean that the degree of coverage of the nickel sponge was close to 100%. Modified porous Ni electrodes were observed with a scanning electron microscope with EDS analysis, making it possible to obtain Pd distribution mapping on the electrode surface. The obtained results are presented in Figure 6.

After the modification, the surface of the nickel sponge was analysed for its chemical composition using the EDS method. It should be noted that the obtained results concern only the chemical composition on the surface of the tested material. The analysis showed the presence of nickel as the primary element in the tested material. Moreover, the presence of small amounts of oxygen on the surface and evenly distributed palladium on the metallic matrix of the nickel sponge was detected. The XRD pattern of the Ni flat and porous electrode decorated by Pd is shown in Figure 7.

Three peaks of Pd at 2θ = 40.5°, 46.8° and 68.4°, corresponding to the (111), (200) and (220) lattice planes, respectively, were observed. All the diffraction peaks can be well indexed to the face-centred cubic (fcc) Pd according to the JCPDS card No. 05-0681, indicating the presence of a Pd crystalline phase. Scherrer’s equation estimates the mean crystalline size to be about 8.27 nm, determined for reflex Pd (111). The intensity of the signals from Pd for flat Ni electrodes was less visible in comparison to the porous substrate due to the much lower concentration of the noble metal on the surface of the analysed material. However, the evolution of the Pd ions during the surface modification can be registered by UV-Vis measurements, presented in Figure 8.

It is well known that the Pd^2+^ chloride complexes absorb light in the UV-Vis range. This property can be used to monitor the process of Pd deposition at the surface of the flat and porous electrode. The obtained results for the flat Ni electrodes are shown in Figure 8. In the UV-Vis spectrum, four peaks were observed. The peak located at 238 nm and originating from the Pd^2+^ chloride complex was used for the kinetic plot. It has to be underlined that obtained UV-Vis spectrum is slightly different from the one presented in the literature [34]. This is probably related to the fact that Pd(II) complex structures are highly dependent on chloride ions concentrations and can also form hydroxy and aqueous complexes [35]. The results connected with porous electrodes are presented in Figure 9. 

The stabilisation of the absorbance value as a function of the modification time, presented on the kinetic plot (Figure 9B), can be attributed to the end of the galvanic displacement process. The metallic nickel and palladium ions exchange was stopped when Pd covered whole Ni surfaces. In both cases, a decrease in the content of Pd^2+^ ions in the electrolyte was visible. This confirms that the reduction reaction of Pd^2+^ ions occurred on the surface of both the flat electrode and the porous electrode. Based on the registered Pd ions concentrations according to microwave plasma atomic emission spectrometer MP-AES, the difference before and after the decoration process for flat (30 min) and porous electrodes (60 min) was 0.11 mM and 0.20 mM, respectively.

### 2.3. Electrochemical Characterisation and Catalytic Activity of Flat and Porous Nickel Electrodes with and without Palladium Modification

Electrochemical measurements can be used to determine the development of the surface area of metallic electrodes. These measurements were performed for flat and porous nickel electrodes, revealing the differences in the electrochemically active surface area (ECSA) [36]. Cyclic voltammetry measurements for a flat nickel sample and a porous sponge in a non-Faradaic region are shown in Figure 10. 

The estimation of the ECSA parameter is based on the electrochemical measurement of the double-layer capacitance. The registered maximum of cathodic and anodic currents scans are plotted as a function of the scan rate, and the obtained slope of the linear function describes the capacitance of the analysed electrode, which is presented in Figure 11.

For ECSA parameter determination, this value has to be divided by the capacitance of an ideal flat surface of the catalyst. The value for the specific capacitances for different materials can be found in the literature. For example, McCrory et al. reported in their benchmarking publications that the Cs values for other catalysts varied between 22 and 130 µF∙cm^−2^ in alkaline solutions [36]. Nevertheless, there are many publications where the average value of 40 µF∙cm^−2^ reported by McCrory has been used as an ideal model for atomically flat electrodes. In both cases, the scans were recorded in the range from 0.01 to 0.2 V∙s^−1^, and the recorded shape was characteristic of the measurements of the double-layer capacity in alkaline solutions. In the case of the tested electrodes, it was observed that despite the identical geometric surface of the samples, the active surface of the sponge (C_dl_ = 1.297 mF) was 5.5-times larger than that of a flat nickel electrode (C_dl_ = 0.2329 mF). Therefore, the obtained parameter can be used to estimate the roughness factor (RF) by dividing the ECSA value by the geometrical area of an examined sample. For the ethanol oxidation, the obtained CV scans were recalculated and normalised to the geometrical feature of the electrode. Moreover, all electrochemical measurements were applied with the Hg/HgO reference electrode to ensure the stability of obtained results. The electrochemical activity of the flat nickel electrode in the ethanol oxidation reaction is presented in Figure 12. 

The shape of the voltametric curves for a flat nickel electrode in a solution without ethanol (black) and its addition at a concentration of 1 (red) and 10 M dm^−3^ (green) were significantly different. The black curve was plotted with no ethanol in the electrolyte for the scan. At a potential of +1.58 V, an anode peak was observed, originating from the oxidation process of the electrode surface. According to Barbosa and colleagues, metallic nickel is automatically covered in strongly alkaline solutions with a thin layer of NiO·H_2_O and Ni(OH)_2_ hydroxide [37], which, when the appropriate potential value is reached, turns into NiOOH (peak C_1_) according to the reaction: Ni(OH)_2_ + OH^−^ = NiOOH + H_2_O + e(1)

Further scanning toward the positive potentials results in the initiation of the water decomposition process with the release of oxygen on the surface of the tested electrode (+1.78 V) marked as A_3_, according to the reaction:2H_2_O = 4e + 4H^+^ + O_2_
(2)

During the return scan, toward negative potential values, a decrease in the current density related to the water decomposition process was observed, which stopped at the potential of +1.74 V. In the other part of the scan, the cathode signal was observed, related to the reversible reaction of the NiOOH phase reduction according to mechanism from the reaction, reaching its maximum at +1.50 V. The recorded peaks remained unchanged, along with the increased number of cycles performed. However, in the case of carrying out scans in the presence of 1 M (red scan), the appearance of a new anode peak was observed, originating from the ethanol oxidation process, starting after exceeding the potential of + 1.62 V and reaching a maximum of 1.71 V. Based on the work of Barbosa and Fleischmann [37,38,39], this process requires pre-oxidation of the surface to the form of NiOOH by reaction (3), which allows the oxidation of ethanol to acetate anions:4 NiOOH + CH_3_CH_2_OH + OH^−^ = 4 Ni(OH)_2_ + CH_3_COO^−^(3)

It should be noted that the mechanism of this particular oxidation is not a one-stage process. Instead, two oxidation reactions overlap, one related to the formation of the NiOOH phase on the surface and the other to ethanol oxidation, starting at full surface coverage and gradually becoming inhibited by its reaction with ethanol. A graphical description of the complete process of ethanol oxidation on the nickel surface was proposed by Barbosa [37]. The course of the curves recorded for the highest concentration of ethanol—10 M dm^−3^ looks different than in previously described cases. The plateau of the current density values associated with ethanol oxidation (+1.88 to +1.96 V) was recorded because the diffusion rate of ethanol and the formed products in the near-electrode interface area controls this process. In the case of complete electrode coverage, the reaction rate will be slowed down. After exceeding the potential of +0.86 V, a decrease in the recorded current density was observed. Its gradual increase and the achievement of more positive potentials resulted from the more intense oxygen evolution on the electrode surface. During the return scans, for the potential of +0.4 V, the appearance of two new anode peaks was observed: A_4_ at +0.78 V and a broad and low-intensity A_5_ peak with a maximum at +1.19 V. These signals were related to the oxidation of the acetate ions (A_4_) present in the electrode space, adsorbed on acetaldehyde, or the single-stage ethanol dehydrogenation reaction (A_5_), according to the reaction mechanism suggested by Barbosa. The weak intensity of these signals was related to the low concentration of oxidation reaction products close to the interface area. Cyclic voltammetry scans for the untreated nickel foams are presented in Figure 13. The shape and registered peaks were very similar to the results obtained for the flat electrode.

The peaks from the NiOOH/Ni(OH)_2_ redox pair were more intense than the flat electrode and were connected with significant surface development of the porous nickel. The process of ethanol oxidation at a concentration of 1 M on porous Ni proceeded similarly to the mechanism described for a flat electrode. Still, with the higher current density, the peaks reached a maximum of +163 mA∙cm^−2^ at a potential of +1.97 V. It should be noted that with each subsequent scan, the maximum current density recorded for this peak decreased, which may be linked to the inhibition of the process due to a gradual reduction of coverage by NiOOH, which is crucial for the process of ethanol oxidation. The scans for the highest ethanol concentration in the electrolyte (10 M) showed the highest current density value in the oxidation process. Therefore, in the porous untreated samples, the A_4_ and A_5_ peaks related to the secondary oxidation of acetaldehyde and other products of ethanol oxidation were not observed. Modification of the Ni surface by other metals, much nobler than nickel, is commonly used for boosting the catalytic performance [18]. Electrochemical tests of ethanol oxidation for different modifications by Pd^2+^ times of Ni foams are presented in Figure 14.

For all of the porous modified Ni samples, the A_2_ peak related to ethanol oxidation was observed, and it was more intense than the clean Ni sample. However, it should be noted that this peak’s intensity increased with a longer modification time. The maximum anodic current density for oxidation of ethanol for each sample was as follows: 160, 225, 294 and 346 mA cm^−2^ without treatment (black), with 15 (red), 30 (green), and 60 min (blue) of modification. In addition, peaks A_4_ and A_5_ connected with oxidation products were recorded for electrodes modified longer than 15 min, increasing significantly with each following scan. It can be explained that the presence of peak A3 at +2.1V is related to the oxygen evolution reaction. The palladium presence on the electrode significantly diminishes the overpotential for this particular reaction, which was reported in other works connected with tailoring the materials for anode in water-splitting cells [40].

Metallic Ni is one of the most suitable elements in the case of methanol oxidation [41] due to the fact that it can produce NiOOH/Ni(OH)_2_ forms on the electrode surface, which are highly active materials in this particular reaction [42]. Furthermore, Ni can generate oxygenated species at higher values of pH and lower potentials than Pd. Therefore, the addition of Ni in the matrix of Pd can refresh Pd active sites by enhancing OH-ads species, increasing ethanol oxidation [43,44]. However, the excess of Ni in the Ni-Pd catalyst can generate large amounts of hydroxyl species that can block the adsorption of new active species, reducing the catalytic activity of the material. That is why we decided to modify the porous nickel surface with palladium via electroless modification.

Based on the presented results, the noble-metal decoration of highly developed nickel surfaces can be a promising way to tailor the electrochemical performance of materials dedicated to ethanol oxidation. Chronoamperometric (CA) measurements for ethanol oxidation for flat and porous electrodes with and without modification are presented in Figure 15.

The data obtained from CA measurements are shown in Figure 15. For comparison, the tests were repeated for flat and porous nickel with and without Pd modification. It should be underlined that, for the current density representation in Figure 15, the development of the porous Ni surface determined by double-layer capacitance was not considered. As can be seen, the modified electrodes had superior activity in the flat and porous versions. During the initial time for the bare Ni electrode, the current density gradually dropped, then remained almost constant around 5 mA∙cm^−2^. Higher initial current density values were observed at the bare porous electrode compared to the flat Ni during the measurement time. For the modified flat electrodes, the current density reached almost constant values around 15 mA∙cm^−2^, which indicates good time stability and high resistance against catalyst poisoning. The highest catalytic activity was observed for porous Ni with palladium modification, reaching the current density of around 55 mA∙cm^−2^, which agrees with previously described results in cyclic voltammetry measurements. The obtained results show great potential for this type of electrode material in ethanol oxidation. 

## 3. Materials and Methods

The following chemicals were purchased and used without any further purification: hydrochloric acid (HCl), sodium hydroxide (NaOH) (99.9%, POCH, Gliwice, Poland), ethanol (96%, Chempur, Piekary Śląskie, Poland), sodium chloride (NaCl, 99.9%, POCH, Gliwice, Poland), potassium chloride (KCl, 98%, Chempur, Piekary Śląskie, Poland), aluminium fluoride (AlF_3_, 99.9, POCH, Gliwice, Poland). All of the used chemicals were analytical grade. 

### 3.1. Preparation of Porous Electrodes

A pure Ni plate was used as the substrate. The sample surface was polished using # 800 emery paper and cleaned using an acetone ultrasonic bath. The electrodeposition of Al formed a Ni aluminide layer. The electrodeposition was performed using molten salt as the medium. The electrolytic bath was an equimolar composition of a NaCl–KCl mixed salt with 3.5 mol% AlF_3_. After the Al deposition, only Al was dissolved to produce a porous surface.

A mixed salt of NaCl–KCl–AgCl (45:45:10 mol%) was placed in a mullite tube having an outer diameter of 6 mm and a length of 500 mm. An Ag wire was then immersed in this salt to serve as the reference electrode. The melt temperature for the Al deposition and Al dissolution was 900 °C, respectively. During the experiment, Ar gas flowed into the cell at 200 cc∙min^−1^. Before forming the porous layer, anodic polarisation curves were measured to investigate the oxidation reaction of the Al ions in the NaCl–KCl molten salt. At this time, the potential sweep speed was 100 mV∙min^−1^. As a result, Al was deposited by constant potential electrolysis with the potential −1.9 V. The dissolution of Al was then performed at −0.5 V. After the treatment, the sample was removed from the bath, and the salt attached to the sample surface was removed by washing it with water. The Pd^2+^ ions concentration in the electrolyte before and after electrode modifications was monitored using a Shimadzu 2501PC spectrophotometer (Kyoto, Japan). This was verified using a microwave plasma atomic emission spectrometer (MP-AES; Agilent 4200, Santa Clara, CA, USA). A standard solution with a Pd concentration of 1000 µg/mL ±7 µg/mL was obtained from SCP Science Solution and diluted appropriately.

The galvanic displacement process was carried out in a standard 3-electrode system with a saturated calomel electrode as a reference, platinum wire as an anode and nickel specimens (flat and porous sample) as a working electrode. The electrolyte in the displacement process consisted of PdCl_2_ dissolved in 0.1 M HCl solution to achieve a concentration equal to 1 mM Pd^2+^ ions. The working electrode’s potential was measured by a Biologic SP150 potentiostat coupled with a personal computer and dedicated software (EC-Lab—Biologic, Seyssinet-Pariset, France).

### 3.2. Physical Characterisation of the Electrodes

The sample’s surface was observed and analysed by scanning electron microscopy (SEM) and X-ray microanalysis (EDS) (JEOL 6000 Plus, Tokyo, Japan). Measurements were performed for bare Ni plates with and without Pd modification and selected porous electrodes. The phase composition was analysed with the XRD method (Rigaku MiniFlex II, Tokyo, Japan) using a copper lamp (λ = 1.54059 nm). The obtained diffractograms ranged from 20° to 90° with a scan speed of 0.5 deg min^−1^. The shape and location of registered signals on diffractograms were compared with characteristic cards from the ICDD database. 

### 3.3. Electrochemical Measurements

The electrochemical investigations were performed in 1 M NaOH solution in a 3-electrode electrochemical cell, similar to the one used for the surface modification. The electrochemically active surface area (ECSA) was estimated through the cyclic voltammetry measurements under a non-Faradaic region as a series of CV scans performed at different scan rates (10, 25, 50, 75, 100, 150, 200 V s^−1^). The registered currents in the anodic and cathodic scans were plotted as a scan rate function according to Equation (4):i = ν · C_dl_(4)
where

i—registered current (mA);ν—applied scan rate (V s^−1^);C_dl_—double layer capacitance (µF cm^−2^).

The slope of the obtained straight line is related to the double-layer capacitance (*C*_dl_) and was used to estimate the surface area determination. The obtained values of C_dl_ for flat and porous electrodes were divided by the value 40 µF∙cm^−2^, which corresponds to the capacitance of the Ni atomic monolayer. The obtained calculation value represents the ECSA parameter, commonly used for normalising the catalytic activity of porous and flat electrodes.

Cyclic voltammetry scans performed catalytic activity measurements in the 1 M NaOH electrolyte with different addition of ethanol, varied from 0 to 10 M dm^−3^. In the case of all experiments, the potential values were recalculated to Reversible Hydrogen Electrode (RHE) potential, according to Equation (5):E(RHE) = E(REF) + pH · 0.059 V + E°(REF)(5)

In addition, catalytic activity in the ethanol oxidation process was performed for flat and porous nickel electrodes without and with surface modification by Pd^2+^ ions at different times and ethanol concentrations.

## 4. Conclusions

This work aimed to develop materials characterised by high catalytic properties for ethanol electrooxidation. Because Raney’s alloy is commonly used as a catalyst, the authors attempted to synthesise Ni-based coatings with a large active surface. As a result, they increased catalytic properties compared to flat Ni electrodes.

The porous nickel electrodes can be used as anodes for the electrochemical oxidation of ethanol in alkaline solutions. Using porous electrodes obtained by selective dissolution in molten salts allows for recording higher current densities than for flat electrodes. Electrochemical studies have shown that a porous material has an electrochemically active surface about 5.5-times larger than a flat electrode. Modifying nickel electrodes by the electroless deposition of palladium from dilute chloride solutions allows for uniform surface coverage. This process is controlled by the diffusion rate of palladium ions into the porous structure. As a result, the electrochemical oxidation process of ethanol occurs on modified electrodes with higher current densities. In cyclic voltammetry scans, the highest catalytic activity was registered for porous Ni-Pd decorated in 60 min, where the maximum oxidation peak for 1 M ethanol achieved +393 mA cm^−2^ compared to the porous unmodified Ni electrode at +152 mA cm^−2^ and flat Ni at +55 mA cm^−2^. In chronoamperometric measurements in ethanol oxidation, the highest activity was recorded for a porous sponge after modification in a solution containing palladium ions, obtaining a current density value of about 55 mA cm^−2^, and for a flat unmodified electrode, only 5 mA cm^−2^ after 1800 s.

## Figures and Tables

**Figure 1 ijms-24-03836-f001:**
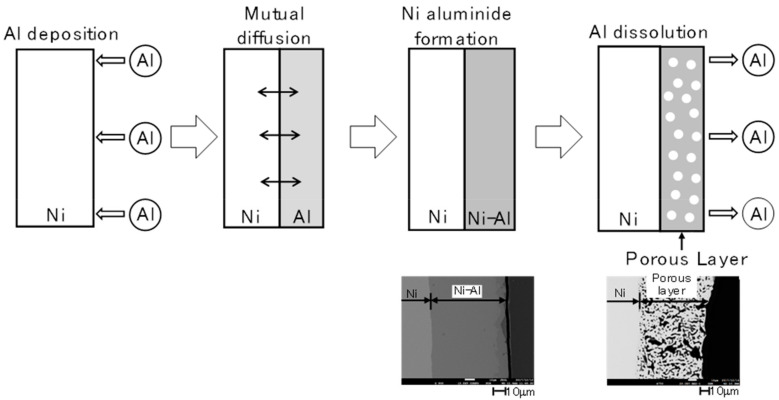
Schematic diagram of the porous layer formation with a cross-section of porous Ni.

**Figure 2 ijms-24-03836-f002:**
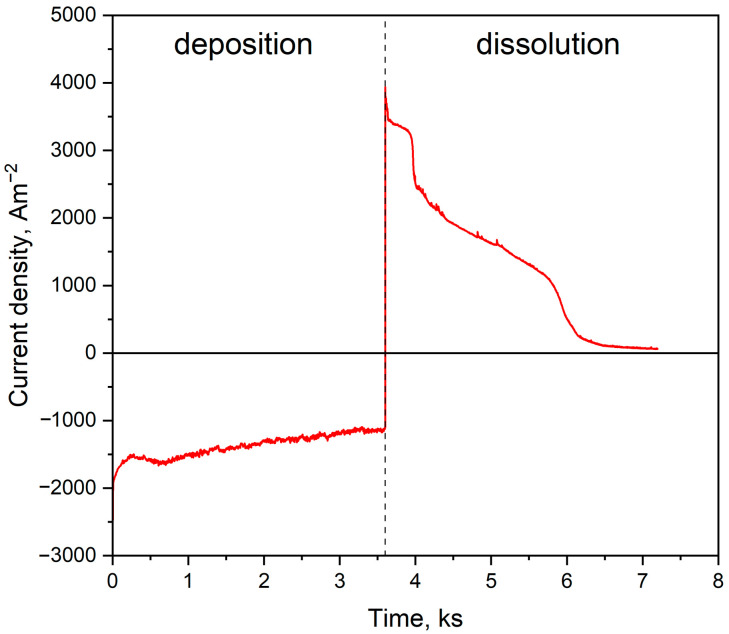
Current density-time curves of the Al deposition at −1.9 V and Al dissolution at −0.5 V in NaCl-KCl-3.5 mol%AlF_3_ melt at 900 °C.

**Figure 3 ijms-24-03836-f003:**
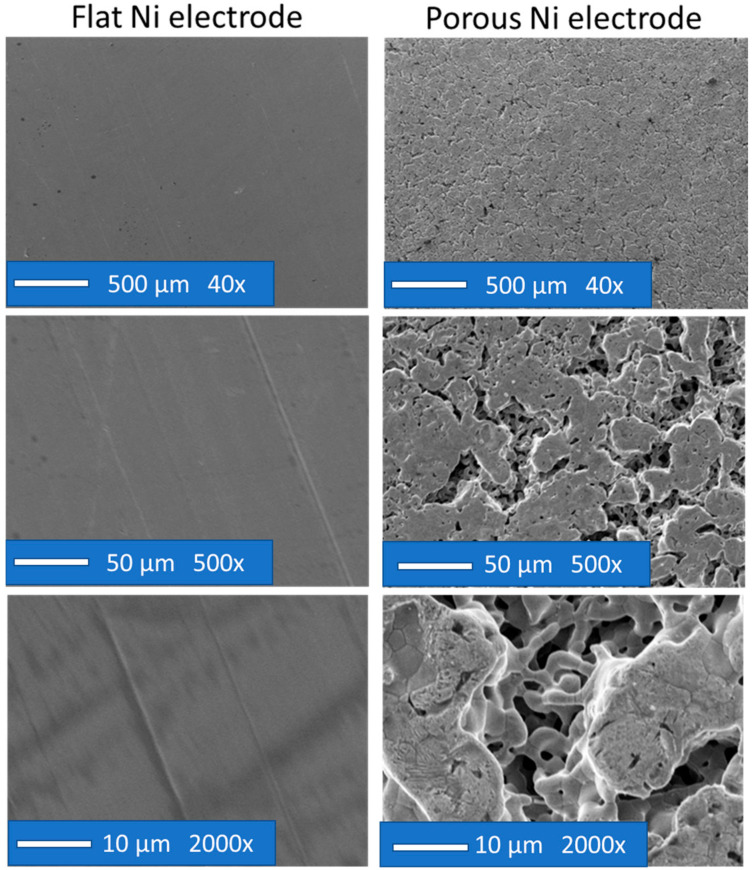
SEM micrographs of the flat Ni sample (left) and porous Ni electrode (right) under different magnifications: 40×, 500× and 2000×.

**Figure 4 ijms-24-03836-f004:**
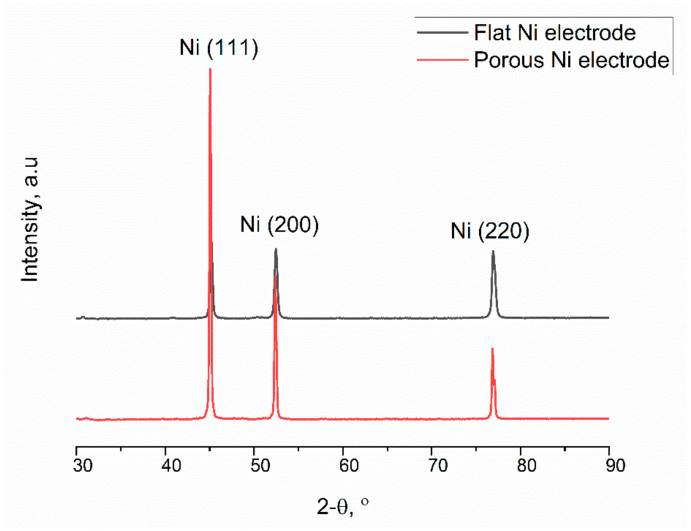
XRD diffraction patterns for the flat Ni substrate (black) and Ni porous electrode (red).

**Figure 5 ijms-24-03836-f005:**
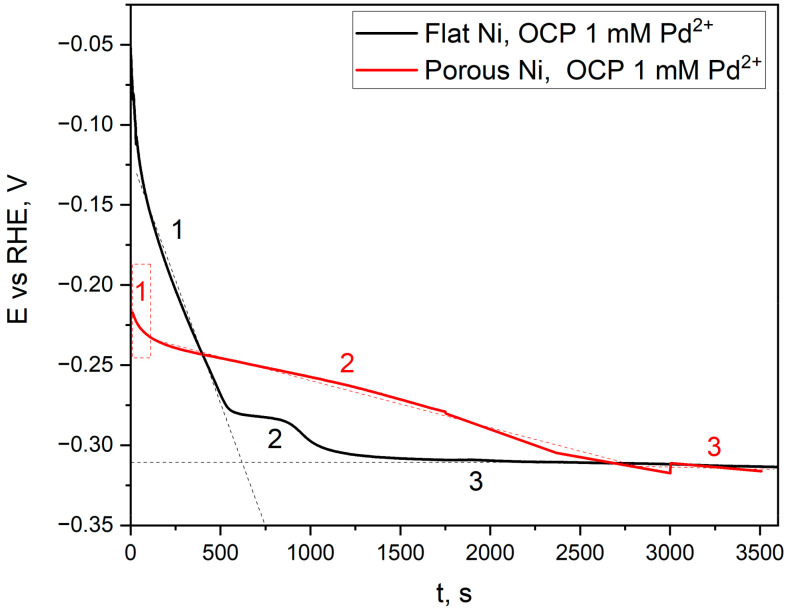
Open-circuit measurements for the flat Ni (black) and porous Ni (red) samples immersed in the 1 mM PdCl_2_ + 0.1 M HCl solution for 3600 s.

**Figure 6 ijms-24-03836-f006:**
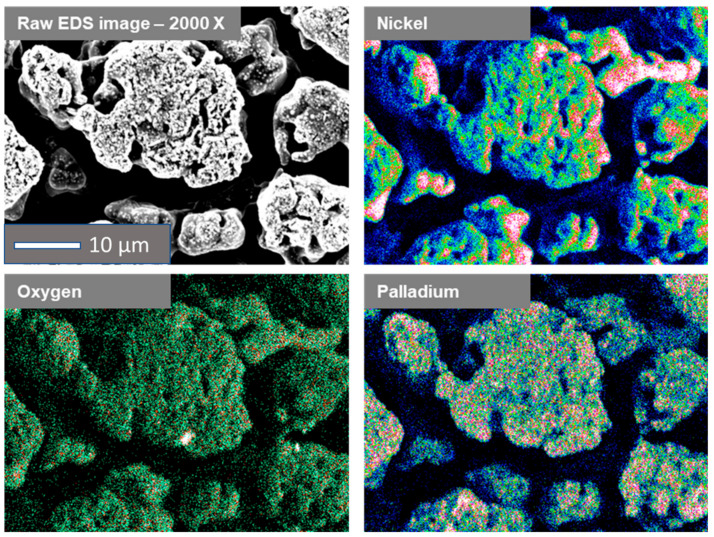
Mapping analysis for a porous Ni electrode decorated by Pd ions in the galvanic displacement process after 3600 s.

**Figure 7 ijms-24-03836-f007:**
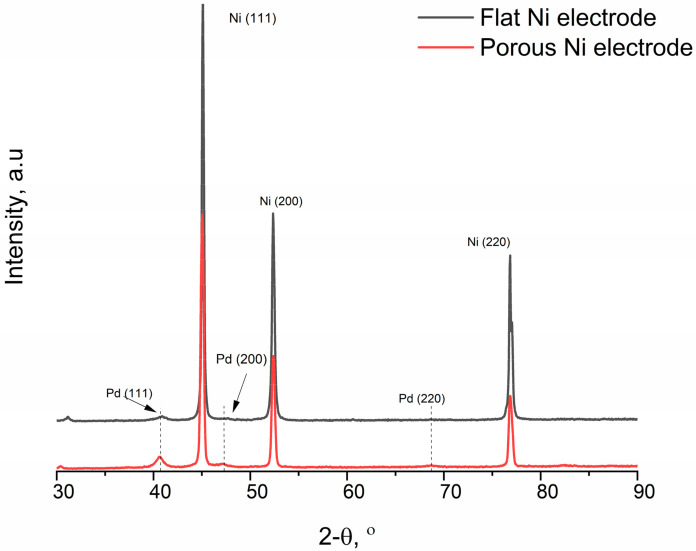
XRD diffraction patterns for flat Ni substrate (black) and Ni porous electrode (red) modified by Pd ions for 3600 s.

**Figure 8 ijms-24-03836-f008:**
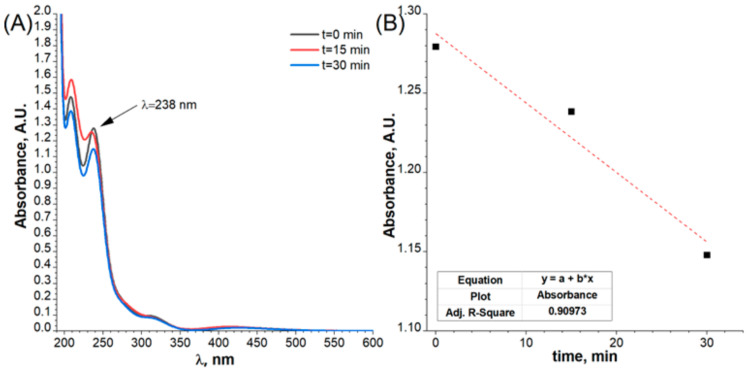
(**A**) UV-Vis spectra of the electrolytes with Pd^2+^ ions after galvanic displacement of a flat electrode with different reaction time and (**B**) kinetic plot for Pd decoration of Ni surface.

**Figure 9 ijms-24-03836-f009:**
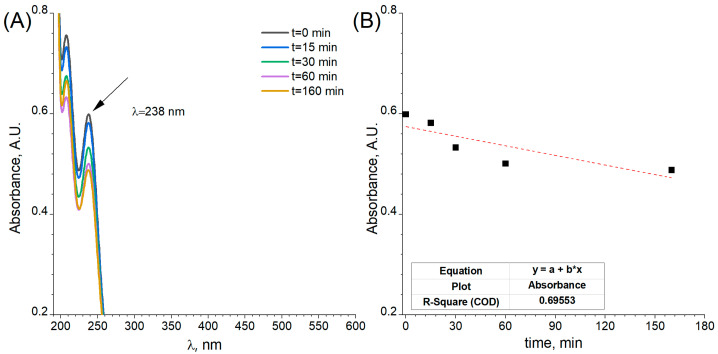
(**A**) UV-Vis spectra of the electrolytes with Pd ions after galvanic displacement of a porous electrode with different reaction time and (**B**) kinetic plot for Pd decoration of Ni foam.

**Figure 10 ijms-24-03836-f010:**
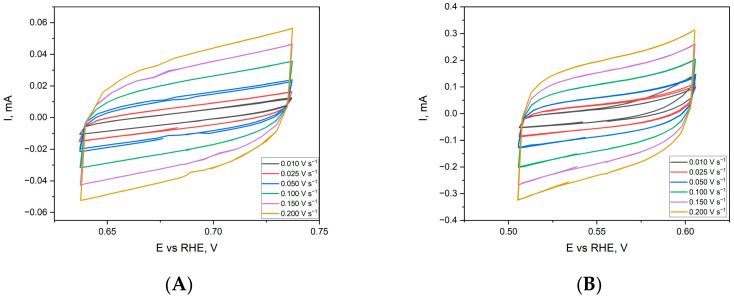
Cyclic voltammetry scans for flat Ni (**A**) and porous Ni sample (**B**) in 1 M NaOH electrolyte in the non-Faradaic potential region. CVs registered for different sweep rates between 0.01 and 0.2 V s^−1^.

**Figure 11 ijms-24-03836-f011:**
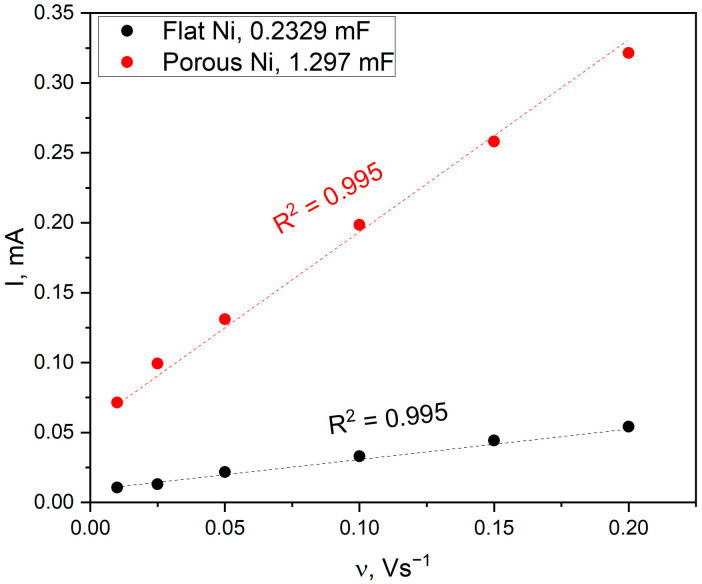
Double-layer capacitance of Ni flat (black) and porous Ni sample (red) determined by cyclic voltammetry measurements in the non-Faradaic potential regions with different sweep rates.

**Figure 12 ijms-24-03836-f012:**
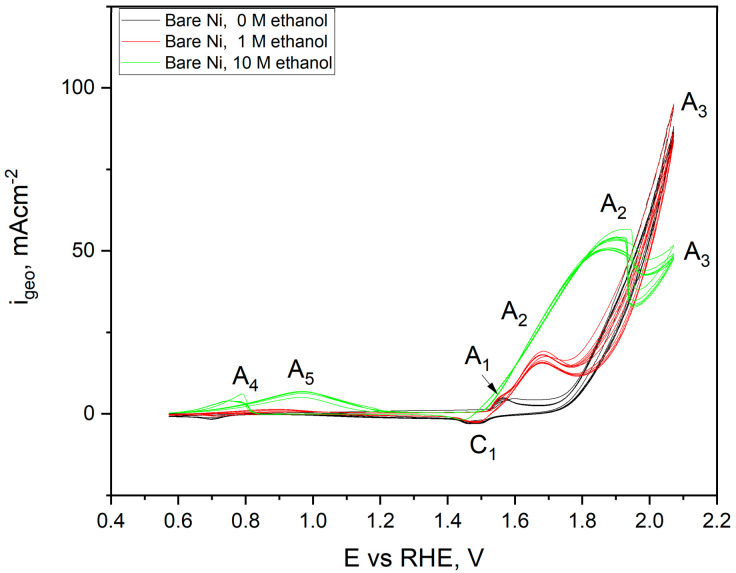
Cyclic voltammetry scans performed for flat Ni electrode in 1 M NaOH electrolyte with and without ethanol addition (1 and 10 M) registered with a sweep rate of 50 mV s^−1^. Ten cycles were performed in one CV scan.

**Figure 13 ijms-24-03836-f013:**
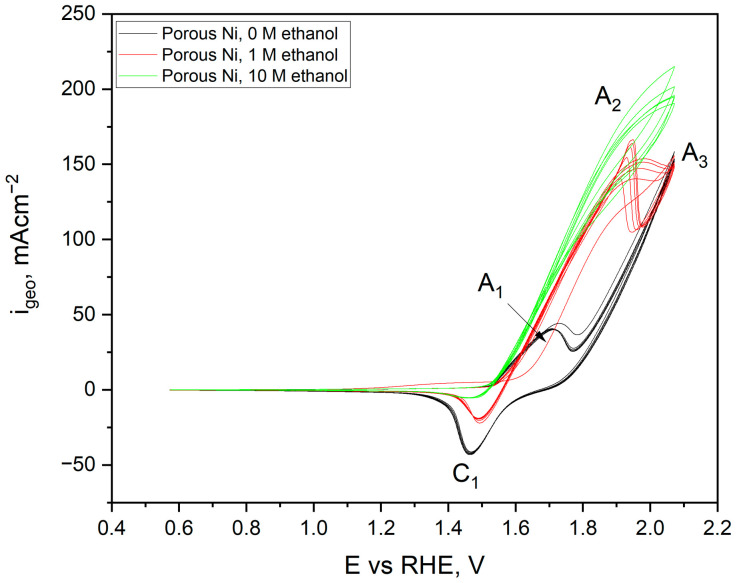
Cyclic voltammetry scans performed for porous Ni electrode in 1 M NaOH electrolyte with and without ethanol addition (1 and 10 M) registered with a sweep rate of 50 mV s^−1^. Ten cycles were performed in one CV scan.

**Figure 14 ijms-24-03836-f014:**
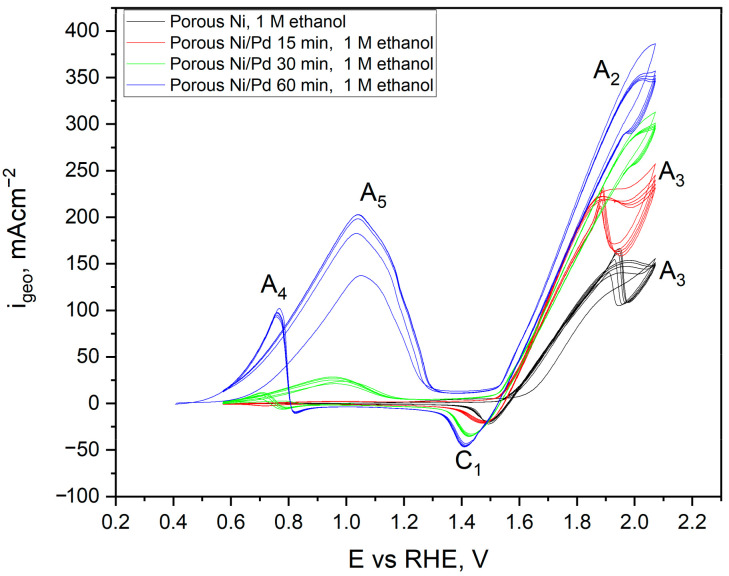
Cyclic voltammetry scans performed for porous Ni and Ni/Pd-decorated electrode in 1 M NaOH electrolyte with 1 M ethanol addition. Pd—decoration time: 0, 15, 30, 60 min. CVs registered with a sweep rate of 50 mV s^−1^. 10 cycles are performed in one CV scan.

**Figure 15 ijms-24-03836-f015:**
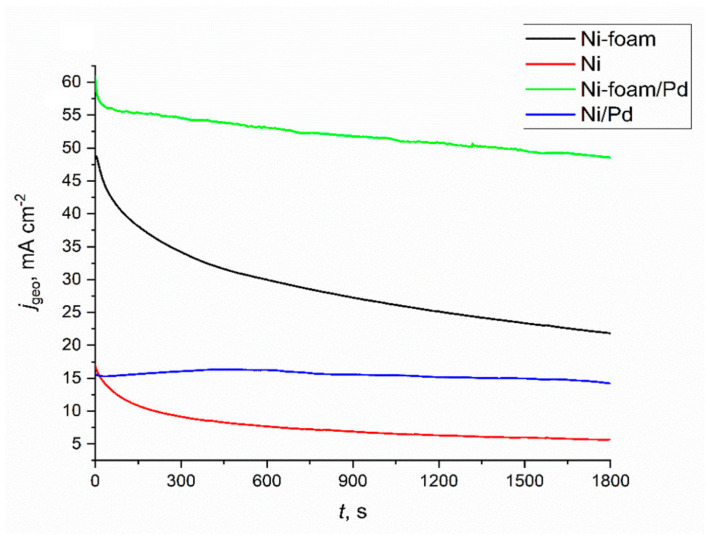
Chronoamperometric measurements in 1 M NaOH + 1 M ethanol addition for flat and porous Ni samples with and without the surface modification with Pd^2+^ ions. Modification time: 3600 s. Applied potential: 1.6 V.

## Data Availability

Data sharing is not applicable.

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
