# Peer review of "Electrocatalytic Performance of Ethanol Oxidation on Ni and Ni/Pd Surface-Decorated Porous Structures Obtained by Molten Salts Deposition/Dissolution of Al-Ni Alloys"

_ijms, 2023, doi:10.3390/ijms24043836_

Round 1
Reviewer 1 Report
The manuscript IJMS-2164223-peer-review-v1 entitled “Electrocatalytic performance of ethanol oxidation on Ni and Ni/Pd surface-decorated porous structures obtained by molten salts deposition/dissolution of Al-Ni alloys” has been evaluated carefully. Authors have synthesized Porous Ni foam electrodes by deposition of Al in NaCl–KCl-3.5 mol%AlF3 molten salt mixture at 900 °C. Flat Ni plates were also compared to obtained porous structures in electrochemical measurements in alkaline solutions. The electrocatalytic activity of flat and porous materials was investigated for ethanol oxidation reaction in an alkaline medium. In my opinion the work is worthable and idea is looking good. It can be accepted for publication in this journal after addressing the questions raised. Authors may improve the manuscript according to the following suggestions to make it more beautiful for publication in this journal.
The author should include below comments in the revised manuscript.
1. The work is attractive from scientific point of view, however, the presentation should be improved. Some of the sentences are confusing/hard to understand by a relatively new person in this field/area of research. The Abstract and conclusions should be made very much attractive and easy to understand for the readers.
2. It is important to add few sentences, in the last part of the introduction section, which highlight the statement of the current research problem. It should focus how this work is better than the previous literature.
3. The experimental part is systematic, however, I would suggest the authors to make the discussion part simpler and more explanatory for academic interest please.
4. Please include 1-2 sentences in conclusion to show the superiority of the present work over the previously published.
5. Please include below references on the topic mentioned in this manuscript and compare the work with the summary of the following papers in brief. (1-2 brief sentences), from materials, catalysis point of view.
1. ““Synthesis and Characterizations of PdNi Carbon Supported Nanomaterials: Studies of Electrocatalytic Activity for Oxygen Reduction in Alkaline Medium” (Molecules, MDPI, Molecules, Volume 26, Issue 11, Pages 1-19 of Article No. 3440, Online 05 June 2021 with Manuscript DOI: https://doi.org/10.3390/molecules26113440 ).
2. “Preparation of Pd-Ni nanoparticles supported on activated carbo for efficient removal of basic blue 3 from water” (Water, MDPI, Volume 13, Issue no. 9, Pages 1-19 of Article no. 1211, Online April 27, 2021 Manuscript DOI; https://doi.org/10.3390/w13091211).
3. Experimental and DFT Studies of Au Deposition Over WO3/g‑C3N4 Z‑Scheme Heterojunction” (Nano-Micro Letters, Volume 12, Issue 1, Article No. 7, Pages 1-18, January 2020 with https://doi.org/10.1007/s40820-019-0345-2).
4. “Catalytic effect of 1,4-dioxane on the kinetics of the oxidation of
iodide by dicyanobis(bipyridine)iron(III) in water” (Catalysts, Volume 11, Issue 7, Pages 1-24 of Article No. 00840 July 11, 2021 with Manuscript DOI: https://doi.org/10.3390/molecules26113440).
Author Response
Answers:
Q) The work is attractive from a scientific point of view. However, the presentation should be improved. Some of the sentences are confusing/hard to understand by a relatively new person in this field/area of research. The Abstract and conclusions should be beautiful and easy to understand for the readers.
A) Abstract has been revised and detailed, and some issues are more precisely explained, especially for new readers in this field. Similar cases have been performed in the case of the result and discussion section, connected with the formation of porous nickel substrate (how this process occurs, explanation of the reactions in the solid state between Ni and Al, and the dissolution process.
Q) It is essential to add a few sentences in the last part of the introduction section that highlight the statement of the current research problem. It should focus on how this work is better than the previous literature.
Suggested sentences connected with the novelty and particular interest of performed research are included in the manuscript.
Q) The experimental part is systematic. However, I would suggest the authors make the discussion part more straightforward and explanatory for academic interest.
Some experimental parts have been revised and simplified to be much more understandable.
Q) Please include 1-2 sentences in conclusion to show the superiority of the present work over the previously published one.
The conclusion part has been modified, and some additional sentences which underline differences or superiority of presented results are included as follows:
Q). Please include the below references on the topic mentioned in this manuscript and compare the work with the summary of the following papers in brief. (1-2 brief sentences), from materials and catalysis points of view.
A). Recommended articles 1 and 2 are connected with Ni-Pd-based materials and are included in the manuscript. Still, a comparison of their properties is not possible because completely different electrochemical reactions are taken into consideration.
Reviewer 2 Report
Manuscript ID: ijms-2164223
Referee Comments: Date: 01/13/2023
This manuscript by Kutyla et al., entitled as “Electrocatalytic performance of ethanol oxidation on Ni and Ni/Pd surface-decorated porous structures obtained by molten salts deposition/dissolution of Al-Ni alloys” presents the chemical synthesis of porous Ni electrode for the electrochemical oxidation of ethanol in alkaline solutions. The porous Ni electrode were synthesized via Al deposition and dissolution method. Further, to enhance the performance, the author introduced palladium ions into the porous Ni structure. Porous Ni-Pd decorated electrode shows maximum oxidation peak for 1 M ethanol is +393 mA cm-2 and higher current density value of 55 mA cm-2. The authors used several characterization techniques to understand the differences between 3 related to their physico-chemical properties. Therefore, we recommend the publication of this paper in International Journal of Molecular Sciences after major revision. Some other issues need to be addressed.
Abstract and Introduction:
1. The overall logic of this manuscript could be improved, such as the introduction.
2. How did you calculate the active surface area?
3. What do you mean by smooth Ni in line 24?
4. Are 5.5-time enhanced surface area shows best performance? Can it be increase or decrease with the dissolution time?
5. Why is there disparity in the peak intensity at t=0, 15 and 30 mins in UV-Vis spectra figure 8.
6. Why the plasmonic peak is visible at t=0 min for Ni electrode in UV-Vis spectra figure 8?
7. Please provide the concentration of Pd decoration on Ni foam?
8. How is this work being novel than the other reported literature? Please provide the comparison table?
9. The author should provide the XPS spectra of all the three cases before and after cycling. You may refer following paper to understand further:
https://doi.org/10.1002/aesr.202100137
10. The author should summarize the result and discussion before the conclusion as well.
Author Response
Dear Reviewer,
we are very thankful for all your suggestions. Below you can find our answers.
Answers:
- The overall logic of this manuscript could be improved, such as the introduction.
Answer: The introduction section have been revised and polished.
- How did you calculate the active surface area?
Answer: The estimation of the “active surface area” have been performed according to electrochemical measurements – Cyclic voltammetry scans in non-faradaic region, where no electrochemical reactions takes place.
According to many publications connected with this topic, this can be done for measurements of intensity of the adsorption/desorption peaks – but this method is limited for some particular materials (for example Pt electrodes). So in our case we performed several CV scans in the non-faradaic potential regime, and registered value of current is connected with double-layer charging/discharging process. CV scans were performed with different scan rate and obtained current were plotted in a function of this parameter. Based on the work (J. Am. Chem. Soc. 2015, 137, 13, 4347–4357 – DOI: 10.1021/ja510442p) the double-layer capacitance of the system is taken as the average of the absolute value of the slope of the linear fits to the data. Moreover, this particular values have been obtained for smooth Ni electrode (0.2329 mF) and porous Ni (1.297mF), which shows that the porous electrode have much more developed surface area (5.57 times bigger).
- What do you mean by smooth Ni in line 24?
Answer: Smooth Ni electrode term is attributed to nickel electrode without modification in high-temperature deposition/dissolution of Al, which makes them porous. That’s flat nickel substrate.
- Are 5.5-time enhanced surface area shows best performance? Can it be increase or decrease with the dissolution time?
Answer: Yes, porous Ni electrode with Pd on the surface exhibit the highest current value in ethanol oxidation reaction. In this particular issue, the dissolution time of Al during high-temperature electrodeposition were not investigated. For this study we used the nickel foams which were fabricated as follows: 3600s of Al deposition at -1.9 V and 3600s of Al dissolution at -0.5 V in NaCl-KCl-3.5 mol%AlF3 melt at 900º C. Different deposition/dissolution time were not investigated here, but in other works of Fukumoto: (DOI: 10.1016/j.ijhydene.2020.07.192).
- Why is there disparity in the peak intensity at t=0, 15 and 30 mins in UV-Vis spectra figure 8.
Answer: For our estimations we are looking on the peak located at 238 nm, due to the fact is connected with presence of Pd2+ complex in electrolyte. Taking into account the Lambert-Beer equation, based on the change in the level of Pd(II) absorbance in the solution, the amount of Pd(0) deposited on the Ni surface was determined.
Abs = ε·l · [Pd(II)]
Where:
ε - molar absorption coefficient
l - optically active path length
[Pd(II)] - concentration of Pd(II) ions in the solution.
- Why the plasmonic peak is visible at t=0 min for Ni electrode in UV-Vis spectra figure 8?
- Please provide the concentration of Pd decoration on Ni foam?
Answer: Presented UV-Vis spectra are not plasmonic peaks of metallic Ni. The UV-vis spectra presents absorption of the radiation by Pd(II) complexes. Absorbance at t=0 corresponds to the initial Pd(II) concentration in the bath used for palladium deposition at Ni surface. Based on the difference between inicial Pd concentration and after modification process we could determine the concentration of Pd on the electrode.
„Based on the registered Pd ions concentrations according to microwave plasma atomic emission spectrometer MP-AES, the difference before and after decoration process for flat (30 min) and porous electrodes (60 min) are 0.11 mM and 0.20 mM respectively.”
- How is this work being novel than the other reported literature? Please provide the comparison table?
Answer: In scientific literature, authors mostly work with nanoparticles tested on electrochemically-conductive substrates. Due to their nature it is much easier to estimate the mass of catalysts – Ni and Pd respectively and their surface area. Moreover, the activity is calculated as the area of the anodic peak connected with ethanol oxidation and divided by the catalyst’s mass. In our example this type of estimation is not possible, due to some factors as follows:
The mass of “active” catalyst cannot be properly calculated. Based on the UV-Vis measurements and plasma atomic emission spectrometer MP-AES, the palladium amount on the surface can be determined, but there are lack of information connected with nickel surface coverage and “free Ni” surface area inside of the porous structure.
Based on that, the comparison with other’s data cannot be done in a realiable way.
- The author should provide the XPS spectra of all the three cases before and after cycling. You may refer following paper to understand further: https://doi.org/10.1002/aesr.202100137.
Answer: For our particular case on the electrode (porous or flat) surface, we have only Ni (unmodified) and Ni with Pd. The presence of Pd has been detected by EDS measurements (Fig .6) and strong reflexes from Pd on the XRD diffractograms (Fig.7). This particular system was not analyzed with XPS method.
- The author should summarize the result and discussion before the conclusion as well.
Answer: This summary has been added to manuscript.
Reviewer 3 Report
In this manuscript, the authors reported the electrocatalytic performance of ethanol oxidation on Ni and Ni/Pd surface-decorated porous structures obtained by molten salts deposition/dissolution of Al-Ni alloys. The research was performed thoroughly. The author provided sufficient data about the material characterization and electrochemical performance. Therefore, I suggest that the manuscript for publication after addressing the following issues.
1. Too many figures are found, which are recommended to be reorganized and combined.
2. What’s production after ethanol oxidation?
3. The potential should be calibrated to reversible hydrogen electrolyte level.
4. What are the roles of Ni and Pd? Please give an in-depth explanation.
5. More structural information after electrochemical reactions should be provided.
6. Some mistakes were found. Please carefully check and revised.
7. The following papers (Applied Catalysis B: Environmental, 2023, 122388, DOI: 10.1016/j.apcatb.2023.122388; Energy & Environmental Materials 2022, e12441 (DOI:10.1002/eem2.12441); Advanced Functional Materials, 2021, 31, 2009779) are recommended to be cited for improving the manuscript.
Author Response
Dear reviewer,
below you can find our answers for your suggestions.
1) Too many figures are found, which are recommended to be reorganized and combined.
Answer: We revised the manuscript, but all of the presented graphs are essential for understanding conducted research.
- What’s production after ethanol oxidation?
Answer: Based on the obtained results we can see that, the polarization of the electrode towards more positive potentials forced to presence of the oxygen evolution reaction, marked in our work as peak A3.
- The potential should be calibrated to reversible hydrogen electrolyte level.
Answer: All potentials were recalculated according to formula: E(RHE) = E(REF) + pH · 0.059 V + E°(REF), where REF are SCE and Hg/HgO electrodes.
- What are the roles of Ni and Pd? Please give an in-depth explanation.
Answer: Metallic Ni is generally speaking one of the most suitable elements in case of methanol oxidation, due to the fact it can produce NiOOH/Ni(OH)2 forms on the electrode surface, which are highly active materials in this particular reaction. Ni can generate oxygenated species at higher values of pH and lower potentials than Pd. Therefore, the addition of Ni in the matrix of Pd can refresh Pd active sites by the enhancement of OH-ads species, increasing the ethanol oxidation. However, the excess of Ni in the Ni-Pd catalyst can generate large amounts of hydroxyl species that can block the adsorption of new active species, reducing the catalytic activity of the material. That is the reason why we decided to modify the nickel porous surface by palladium via electroless modification.
- More structural information after electrochemical reactions should be provided.
Answer: I would like to apologize, but I do not understand what we should provided.
- Some mistakes were found. Please carefully check and revised.
The manuscript has been detailed reviewed and checked by spelling software.
Reviewer 4 Report
Review IJMS-2164223
The manuscript describes the preparation of porous Ni foam electrodes. The electrocatalytic activity of the electrodes were evaluated for ethanol oxidation reaction in the alkaline medium. The current work could be an interest to the reader of IJMS. The following suggestions were offered to the author for the improvement of the manuscript:
1. Suggest spelling out the novelties of current study and the gap in the research it fills.
2. Kindly provide the instrumentation detail for SEM, EDX, and electrochemical station used.
3. Only 4 out of 28 references used were published in the last 3 years. Suggest doing a more thorough literature review to include a more recent published work.
4. Kindly provide a cross sectional SEM micrograph to prove the porous layer formation as suggested in Fig. 1.
5. Kindly provide scale bar for Fig 6.
6. Kindly scale the data presented in Fig 7. The current presentation is misleading.
7. Kindly provide an error bar for data presented in Figs. 8b and 9b alongside with the kinetic parameter.
8. Kindly report the data presented in Figs. 10, 12 and 13 in E(RHE) (the x-axis) for easy comparisons with other result.
9. Kindly provide error bars for data presented in Fig. 11.
10. Suggest comparing the obtained result with other recently published data.
Author Response
Dear Reviewer,
Below you can find the answers for your suggestions. We are thankfull for your effort to improve the quality of manuscript.
- Suggest spelling out the novelties of the current study and the gap in the research it fills.
Answer: The novelty of the current study is related to preparation and evaluation of Ni-based new types of electrode materials dedicated for electrochemical reactions. Based on the literature review, researchers focus on nanomaterials/nanoparticles like core-shell structures which are later sputtered onto conductive substrate. In our approach we wanted to prepare solid electrodes in a form of metallic sample, with enlarged surface area. The phenomenon of intermetallic phases formation by „itself” in high-temperature mollten salts are not fully examined and there are many factors to investigate.
- Kindly provide the instrumentation detail for SEM, EDX, and electrochemical stations used.
Answer: These details are included in the manuscript.
- Only 4 out of 28 references used were published in the last 3 years. Suggest doing a more thorough literature review to include a more recent published work.
Answer: The citation of some more recent articles has been added to the manuscript.
- Kindly provide a cross-sectional SEM micrograph to prove the porous layer formation as suggested in Fig. 1.
Answer: The cross-section of the porous layer is included in Fig. 1.
- Kindly provide a scale bar for Fig 6.
Answer: Bar scale now is added.
- Kindly scale the data presented in Fig 7. The current presentation is misleading.
Answer: The XRD data are scaled. The graph for non-modified electrode is only shifted slightly to not overlap with the results for modified sample.
- Kindly provide an error bar for data presented in Figs. 8b and 9b alongside with the kinetic parameter.
Answer: Obtained results were taken from UV-Vis spectra. The spectra aquiring were repeated till no changes were observed, so the error bars would be very small and missing on the graph.
- Kindly report the data presented in Figs. 10, 12 and 13 in E(RHE) (the x-axis) for easy comparisons with other result.
Answer: All the experiments in 1 M NaOH-based electrolyte were
- Kindly provide error bars for data presented in Fig. 11.
Answer: The values for Fig. 11 were taken from the maximum of cathodic and anodic currents from Fig.10. As you can see, the values are visually constant, so error bars would be non-visible.
- Suggest comparing the obtained result with other recently published data.
Answer: In scientific literature, authors mostly work with nanoparticles tested on electrochemically-conductive substrates. Due to their nature it is much easier to estimate the mass of catalysts – Ni and Pd respectively and their surface area. Moreover, the activity is calculated as the area of the anodic peak connected with ethanol oxidation and it is divided by the mass of the catalyst. In our example this type of estimation is not possible, due to some factors as follows:
The mass of “active” catalyst cannot be properly calculated. Based on the UV-Vis measurements and plasma atomic emission spectrometer MP-AES, the palladium amount on the surface can be determined, but there are lack of information connected with nickel surface coverage and “free Ni” surface area inside of the porous structure.
Based on that, the comparison with other’s data cannot be done in a realiable way.
Round 2
Reviewer 4 Report
The authors have not made significant improvement based on the comment given. Kindly look through the earlier comments.
Author Response
Dear Reviewer,
We are very thankful for all suggestions and comments connected with our manuscript.
In attachment we included the answers for all your comments.
We strongly believe that it meets your requirements.
Best regards,

Round 3
Reviewer 4 Report
Thank you for revising the manuscript. I trust that the manuscript is now suitable for publication at IJMS.